# Life Quality of Children Affected by Cleft Lip Palate and Alveolus (CLPA)

**DOI:** 10.3390/children9050757

**Published:** 2022-05-21

**Authors:** Marco Pasini, Irene Cagidiaco, Eleonora Fambrini, Marco Miceli, Elisabetta Carli

**Affiliations:** Unit of Pediatric Dentistry, Department of Surgical, Medical, Molecular and Critical Area Pathology, University of Pisa, 56126 Pisa, Italy; cagidiacoirene@gmail.com (I.C.); eleonorafambrini@gmail.com (E.F.); marcoigdmiceli@gmail.com (M.M.); carlibetta@libero.it (E.C.)

**Keywords:** cleft lip/palate, quality of life, OHIP-14

## Abstract

The aim of this study was to investigate the quality of life of children and adolescents affected by cleft lip, palate and alveolus (CLPA) in the Italian population after a tailored treatment dental program. A prospective study was conducted with subjects of both genders at the University Hospital of Pisa, Italy. This study included 32 patients (11 females and 21 males, mean age: 9.8 ± 2.1 years old) affected by CLPA (test group); a tailored oral hygiene instruction protocol was adopted in the test group since early age (6.1 ± 0.9 years old) after corrective surgery and regular dental hygiene appointments were scheduled. Moreover, a control group of 32 patients (11 females and 21 males, mean age: 10.1 ± 2.2 years old) without CLPA was included; control subjects were first-visit patients, matched for age and sex, who had not received a specific dental hygiene program before. The OHIP-14 (Oral Health Impact Profile) questionnaire was applied for the evaluation of the quality of life of patients of both groups and the questions were presented directly to the patients. Moreover, the DMFT (Decayed, Missed and Filled Teeth) index was measured. A statistical analysis was performed and the level of significance was set at: *p* < 0.05. The OHIP-14 scores presented significant differences between the two groups (*p* < 0.05); the test group exhibited a lower mean OHIP-14 score in comparison to the control group. The DMFT score was significantly lower in the test group in comparison to controls (*p*: 0.001, *p* < 0.05). The quality of life and DMFT score of children and adolescents affected by CLPA, after a tailored treatment dental program, were better than that of the subjects of the control group.

## 1. Introduction

Cleft lip and/or palate is the most common malformation of the craniofacial district, which is characterized by the presence of a cleft occurring due to a lack of fusion of symmetrical structures of the face, in particular: of the lip of the alveolar process and palate [1]. The prevalence is about 1–5/10,000 children [2]. The approach to the patient must therefore be multidisciplinary and timely, as the involved anatomical structures can be responsible for problems related to breathing, chewing, swallowing, hearing, phonation and sucking, which are all functions essential to life [3]. Also, the dentist is involved in the treatment of cleft patients, as the prevalence of dental anomalies and subsequent malocclusion is much higher than in the general population [4,5]. Therefore, orthodontic and restorative treatments are often necessary [6,7,8,9,10]. 

Craniofacial anomalies are more prevalent in cleft patients than in non-cleft subjects; Akay et al. (2020) observed that the inter-clinoid distance was statistically lower in cleft patients than in non-cleft individuals [11]. 

Doğan et al. (2021), compared upper and lower maxilla of subjects with unilateral cleft lip and palate to skeletal Class I and Class III patients and found that maxillary development in cleft patients is similar to skeletal Class III [12]. Therefore, orthodontists should consider an upper maxillary protraction and expansion in the early period (4–10 years old).

Poor oral hygiene is often recorded in children with cleft lip and palate, therefore early preventive advices, including toothbrush techniques and healthy diet, should be given by dentists and dental hygienists. Attention should be given to toothbrushing in the cleft region with a soft toothbrush and dental development in the cleft area should be monitored constantly, as stated by Rivkin et al. (2000) [13]. 

Proper oral health in children may improve quality of life. Until recently, only few valid tools were available to evaluate the functional and psychological outcomes of oral disorders. First of all, a distinction has been introduced between perception of the actual oral health status and quality of life related to oral health (Oral Health Related Quality of Life, OHRQoL), the latter understood as a wider interaction between oral cavity health, general health and related quality of life [14,15]. Among the numerous measuring instruments concerning perception of oral health status and OHRQoL, the OHIP questionnaire, which is available in two versions, was frequently adopted. OHIP-14 is the short 14-item form of a 49-item questionnaire, OHIP-49 [15,16], which was developed by Locker and Slade and based on Locker’s conceptual model for oral health. It includes seven domains: functional limitation, physical pain, psychological distress, physical disability, psychological disability, social disability, and handicap [17]. The original English version of the OHIP questionnaire was translated and adapted to the Italian version through a translation and back-translation process (“forward/backward translation”) and then a pre-test for administering the questionnaire was developed. Many in-depth studies on the use of OHIP-14 in adult patients without facial cleft were carried out, whereas only few studies were conducted in patients with cleft lip and palate. Studies applying the OHIP-14 questionnaire for evaluating the quality of life in children and young adults with cleft lip and palate are still lacking; Piombino et al. (2014) validated a quality of life cleft questionnaire in adolescent patients and found that cleft subjects have more problems with self-esteem and social skills [18].

Corcoran et al. (2020) evaluated OHRQoL in young adults with cleft lip and palate and found that more than half of the subjects exhibited an impact on OHRQoL, with an OHIP-14 score ≥ 3; physical pain and psychological discomfort were more prevalent in cleft patients in comparison to controls [19].

Therefore, a tailored dental program should be considered in early age, after the surgical treatment, in order to decrease gingival bleeding and dental caries, improving patient’s quality of life.

The aim of the present study was to evaluate the life quality of cleft/lip palate children and young adolescents after a tailored treatment dental program. The null hypothesis is that there are no differences in OHRQoL between cleft subjects and non-clefts individuals and that the tailored dental program is not effective in improving oral health quality of life.

## 2. Materials and Methods

The present prospective monocentric study was conducted at the University of Pisa (Italy). Patients with right and left CLPA, who had undergone surgical and orthodontic treatment since 2005, were recruited from The Cleft and Palate Center, within the Plastic and Reconstructive Surgery Unit of the Santa Chiara Hospital, in Pisa; furthermore, a dental clinical examination was performed, and interviews were conducted in the Pediatric Dentistry Unit. Dental examinations were performed by two operators (paediatric dentists with more than 10 years of experience in this field) with the same tools and in the same place.

All procedures were conducted in accordance with the principles of the Declaration of Helsinki and informed consents were obtained from the parents before the beginning of the study. The local Ethics Committee approved this study (Tuscany Regional Pediatric Ethics Committee, n. 33/18, 19 March 2018). The study was based on an estimated sample size of 32 subjects, with a ratio of 1:1 for test and controls, which has been calculated to be adequate to achieve 80% power with an alpha error of 0.05% to detect a difference of 5% of proportions of OHIP, ≤2 between test and controls.

Inclusion criteria: patients with unilateral or bilateral CLPA, mean age: 10 ± 2.4 years old, who have undergone a surgical visit or who are under orthodontic treatment. The patients must have already undergone initial corrective surgery in early age. Patients of any nationality, gender, ethnicity, who had orthopantomography and lateral cephalometric radiograph available were included. 

Exclusion criteria: syndromic patients, patients with other craniofacial malformations, patients under the age of 6 and/or older than 15 were excluded. Patients who did not have radiographic examinations such as orthopantomography and lateral cephalometric radiograph were also excluded. Patients who did not give their consent to the execution of the questionnaire, or patients whose parents did not give consent, were excluded. 

A total of 40 subjects (11 females and 21 males), affected by different degrees of CLPA out of 90 patients treated in the Plastic and Reconstructive Surgery Unit met the inclusion criteria and were eligible for this study; 32 participants completed the questionnaire (participation rate: 80%) and were included in the test group.

The examined subjects presented:14 patients: left CLPA (4 females, 10 males)8 patients: bilateral complete CLPA (2 females, 6 males)10 patients: right CLPA (5 females, 5 males)

During the first visit, the patients or their parents were informed about the study, asking for their consent for the submission to a quality-of-life questionnaire (OHIP-14) and with their help the medical record was completed. The 14 questions were presented directly to the patients, and they were asked to answer each question with a score from 0 to 4 (0 = never, 1 = almost never, 2 = sometimes, 3 = quite often, 4 = very much often), based on their experience (Table 1).

Control subjects were randomly selected from the patients referred to the Pediatric Dentistry clinic of the University of Pisa for a dental examination in 2019. A random computerized analysis was carried out in order to select 32 patients, matched for age and sex to the test group, from a pool of 120 eligible children. The control group was made up of 11 females and 21 males, mean age: 10 ± 2.1 years old, that filled OHIP questions at the end of the dental examination.

Each patient was asked to answer the following questions regarding his or her experience in the last 3 months.

A professional dental hygiene treatment, including professionally cleaning of hard and soft plaque using ultrasonic scalers, calculus removal and teeth polish, was performed in CLPA subjects aged 6 years old or more, every 6 months. A Tailored Brushing Method (TBM) was adopted for test group patients, choosing the most suitable dental tools and brushing techniques considering patient’s manual skill and personality profile [20]. Control patients did not receive a specific dental hygiene program before, as stated during the screening.

Decayed, missing, and filled teeth (DMFT and dmft for primary teeth) index was calculated for each patient of the two groups using a dental explorer; measurements were performed by two operators and the mean value of the two measurements was adopted. DMFT index consists of the sum of the number of decayed, missing due to carious disease and filled teeth and represents the current predominant population-based parameter to measure caries experience, also in growing patients as performed in this study.

## 3. Statistical Analysis

The quantitative variables were expressed as mean ± standard deviation. Quality of life, measurement with OHIP questionnaire, and caries status, measurement using DMFT index, were defined as dependent variables while dental hygiene treatment and instructions were defined as independent variables.

The U Mann-Whitney test for nonparametric data and Kruskal–Walis test was used to evaluate whether there were significant differences between the test group and the control group in the scores detected by the OHIP-14 questionnaire and DMFT. The level of significance was set at *p* < 0.05. The statistical analysis was carried out using the SPSS 16.0 software (SPSS, Chicago, IL, USA). 

DMFT was measured by two different operators in order to measure inter-observer variability that was calculated by using Cohen’s Kappa coefficient. The coefficients obtained ranged between 0.91 and 0.97.

## 4. Results

All patients included in the present study answered to each question of the OHIP-14.

The results of this study showed that both test and control group had a high quality of life score; in fact, the mean OHIP-14 values observed for each question at the end of the study were low in both groups.

Significant differences (*p* < 0.05) were observed between the two groups according to the OHIP-14 questionnaire. 

The means and standard deviations of the scores attributed in each group are shown in Table 2. Statistical analysis showed a significant difference between the test group and the control group for question 1 of the OHIP-14 questionnaires; the test group reported a mean lower toothache in comparison to the control group (*p*: 0.004). Moreover, the test group showed lower values for mouth breathing and snoring and for tooth stains; however, the difference between the two groups was not significant (*p*: 0.51).

The control group showed a significantly (*p*: 0.03) lower score in comparison to CLPA patients for broken teeth that create spaces between teeth while reporting a significantly (*p*: 0.047) higher score for recurrent mouth sores in comparison to controls.

CLPA subjects showed lower gingival bleeding in comparison to control patients and the difference was significant (*p*: 0.001). Instead, test patients exhibited slightly higher halitosis in comparison to controls; however, no significant difference was recorded for this question (*p*: 0.72).

The test group exhibited a significantly lower score (*p*: 0.001) for food stuck between their teeth in comparison to control group.

A slightly lower score was observed for tooth pain or sensitivity in CLPA patients in comparison to controls; however, no significant difference was recorded (*p*: 0.098).

Dry lips and mouth scores were slightly higher in the test group in comparison to control subjects, but no significant differences were observed (*p*: 0.09).

Chewing difficulties were slightly lower in CLPA patients in comparison to healthy subjects but the difference between the groups was not significant (*p*: 0.34).

Similar scores were recorded for feeling upset for teeth problems (*p*: 0.42) and for missing school days (*p*: 0.2) in both groups and no significant differences were recorded.

Furthermore, CLPA patients showed a slightly lower score for being reassured about teeth problems in comparison to control subject, but no significant differences were observed (*p*: 0.71).

DMFT was significantly higher (*p*: 0.001, *p* < 0.05) in the control group (3.3 ± 2.4) in comparison to the test group (1.4 ± 0.9). 

## 5. Discussion

Cleft lip and or palate is a complex craniofacial malformation which determines multiple anatomical and functional alterations of the craniofacial complex, providing peculiar facial appearance, speech and communication problems, chewing and respiration patterns alterations that affect quality of life. Cleft children from the very beginning of their life undergo medical checks by a multidisciplinary team of different medical professionals, including dentists in order to improve oral health. The results of the present study showed that a specific dental hygiene program significantly improved oral health related quality of life in cleft children and decreased caries experience; statistically significant differences were highlighted in 5 out of 14 questions of the OHIP-14 questionnaire that is closely related to the health of the oral cavity. In the literature, studies that investigate the quality of life in children and adults with cleft lip or palate were performed.

In a study of Palmeiro et al. [21], the quality of life of adult patients with cleft lip and palate, who had already received surgical treatment and edentulous patients, who lost teeth due to carious processes and periodontal disease was assessed. The authors did not detect substantial differences between the two groups as regards the scores attributed to the questions of the OHIP-14 questionnaire. Both groups reported difficulties in chewing and a worse psychological and functional impact compared to the control group, formed by healthy adult patients. 

Foo et al. [22] compared, by means of the OHIP-14 questionnaire, the quality of life of adult patients with cleft lip and palate with that of healthy adult patients and their results showed that the surgical correction of the malformation did not improve either the quality of life, or the impact on oral health of the cleft lip and palate patients. In both previously mentioned studies, the test group was made up of adult patients and it may be hypothesized that in an adult subject the degree of awareness of his or her own malformation, as well as the impact that this defect could have on both relational and medical spheres, might be different from those of a child. Moreover, in a study of Rando et al. (2018) the oral health-related quality of life in children affected by cleft lip and/or palate has been assessed by the use of a questionnaire. The result of this study highlighted the correlation between the presence of a malformation and a poor quality of life, concerning oral health [23]. In fact, children unaffected by cleft lip and/or palate showed a better quality of life than children with CLP. The main difference between this study and our experience lies in the fact that in our study a specific dental hygiene program was performed in children.

Naros et al. (2018) investigated the oral health-related quality of life in a sample of cleft and non-cleft pediatric subjects, including their parents reporting using the age-specific German KINDL^R^ questionnaire, and found a significantly higher quality of life score in cleft children compared with normative data, similar to our findings. The authors stated that multiple medical professionals support, both from a psychological and from a speech therapy approach, increased self-esteem in children, communication skills and family interaction [24]. 

Similar results were found in our study because patients with cleft lip and palate were subjected to periodic multi-specialist and dental check-ups that improved patients’ quality of life. 

Study limitations included children’s personal reports that could not fully detect patients’ quality of life. Furthermore, respondents may lie due to social desirability or because they had difficulties in understanding and interpretating the questions.

An additional limitation of the present study was the fact that each participant was asked to answer the questions regarding his or her experience in the last 3 months; a wider time range could have affected the answers and the quality of life scores. 

## 6. Conclusions

According to the results of the present study, it was found that the OHIP-14 scores presented significant differences between the two groups and the null hypothesis was rejected; the CLPA group showed a lower mean OHIP-14 score in comparison to the control group. Moreover, the DMFT score was significantly lower in the test group in comparison to controls.

The quality of life and DMFT index scores of children and adolescents affected by CLPA, after a tailored treatment dental program, were better than that of healthy subjects.

The clinical relevance of such results is huge, defining the need of improvement of oral hygiene regimen protocol in patients with CLPA. The complexity of the malformation itself requires a multidisciplinary approach to minimize complications and discomforts for the paediatric patients and the paediatric dentist must be part of the cleft- multidisciplinary team. Moreover, the often-severe malocclusions these patients present requires complex and early orthodontic treatment, for which tooth health is necessary, as they represent the support for any orthodontic appliance [6,7,8,9,10]. In addition, the higher prevalence of tooth anomalies in shape, number, and enamel composition in CLPA patients determine a higher risk for severe carious lesions, also in primary dentition [4,5]. 

Future studies may focus on CLPA quality of life, comparing CLPA patients who underwent from early age a tailored protocol of oral health maintenance and subjects affected by the same malformation who did not receive the same oral hygiene instruction and treatment. An important secondary parameter to be assessed would be the therapy impact of poor oral health and its effects both on surgical and orthodontic prognosis and results.

In conclusion, in this perspective the oral health maintenance in CLPA subjects is indispensable for many reasons and a tailored treatment dental program from early age should be performed. National Health Care Systems should provide preventive protocols in these special-needs patients.

## Figures and Tables

**Table 1 children-09-00757-t001:** OHIP-14 questionnaire specifics.

OHIP-14 Questionnaire Question Number	Score
*(1)* *Do you have toothache?*	0–4
*(2)* *Do you breath through your mouth or do you snore?*	0–4
*(3)* *Do you have stains on your teeth?*	0–4
*(4)* *Do you have broken teeth that create spaces between your teeth?*	0–4
*(5)* *Do you suffer from recurrent mouth sores?*	0–4
*(6)* *Do you have halitosis?*	0–4
*(7)* *Do you have gingival bleeding?*	0–4
*(8)* *Does food stick between your teeth?*	0–4
*(9)* *Do you have pain or sensitivity in your teeth after hot/cold stimulation?*	0–4
*(10)* *Do you have dry lips/mouth?*	0–4
*(11)* *Do you have difficulties biting or chewing?*	0–4
*(12)* *Do you feel upset about your teeth problems?*	0–4
*(13)* *Did you miss school days due to your teeth problems?*	0–4
*(14)* *Were you reassured about your teeth problems?*	0–4

**Table 2 children-09-00757-t002:** Means and standard deviations of the scores attributed in the OHIP-14 questionnaire in the test group and in the control group.

Question Number	Question Topic	*p* Value	Test Group (*N* = 32) Mean (Standard Deviation)	Control Group (*N* = 32) Mean (Standard Deviation)
1	Toothache	0.004 *	0.29 (0.61)	0.81 (0.78)
2	Snoring and mouth breathing	0.51	0.81 (1.4)	1.02 (1.13)
3	Teeth stains	0.31	0.15 (0.51)	0.28 (0.5)
4	Broken teeth	0.03 *	0.9 (0.6)	0.34 (1.25)
5	Mouth sore	0.047 *	0.19 (0.33)	0.49 (0.77)
6	Halitosis	0.72	0.83 (1.16)	0.72 (1.3)
7	Gingival bleeding	0.001 *	0.35 (0.77)	1.44 (0.99)
8	Food sticking	0.001 *	0.87 (0.92)	1.39 (0.79)
9	Sensitivity	0.098	0.8 (1.31)	1.29 (1)
10	Dry lips/mouth	0.09	1.59 (1.36)	1.08 (0.95)
11	Biting and chewing difficulties	0.34	0.19 (0.57)	0.35 (0.76)
12	Dental concerns	0.42	0.4 (1.1)	0.22 (0.61)
13	Missing school	0.2	0.45 (0.95)	0.78 (1.1)
14	Reassurance	0.71	3.76 (0.88)	3.83 (0.6)

U Mann-Whitney and and Kruskal–Walis test. * *p* < 0.05.

## Data Availability

Data is contained within the article.

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
