# Peer review of "Life Quality of Children Affected by Cleft Lip Palate and Alveolus (CLPA)"

_children, 2022, doi:10.3390/children9050757_

Round 1

Reviewer 1 Report

Journal: Children Manuscript ID: children-1739824 Type of manuscript: Research Article Title: LIFE QUALITY OF CHILDREN AFFECTED BY CLEFT LIP PALATE AND ALVEOLUS (CLPA)

This manuscript was intended to investigate prospectively the quality of life of children and adolescents with cleft lip, palate and alveolus (CLPA) after a tailored treatment dental program in Italy. The study used OHIP-14 score and DMFT score in comparison to control group. The study proved that the quality of life and DMFT index score of children and adolescents affected by CLPA, after a tailored treatment dental program implemented there, were better than that of healthy subjects.

I was glad to read this concised and well written article.

Only one remark:

  • The abbreviation ‘DMFT’ should be explained at the first usage in the text including the abstract.

Author Response

Dear reviewers, thank you for your kind and smart recommendations.  Reviewer 1: we added DMFT acronym explanation both in the abstract and in the main text (the very end of Materials and Methods section). 

Reviewer 2 Report

This is an interesting manuscript, adding important information to the field of cleft lip and palate surgery and treament. 

Things that can be improved (and should be improved I think) to make the manuscript easier to read include: 

  • RESULTS: p- values in the text - please use to actual p-value
  • Table 2: I believe that it would be easier to know to which questions the differences detected were referring to by adding a column next to the numbers : i.e. Question 1, "toothache", Question 2, mouth breathing/snoring etc. 
  • Conclusion: That's great that children with CLP undergoing tailored dental treatment showed a better quality of life in comparison to control cohort. However, what does this imply? Do you recommend that all patients with CLP should undergo such treatment?   

Author Response

Reviewer 2: •    We corrected the Results section adding the “p values”, which were reported in the Table 2, in the main text for every single question. •    We added a column in Table 2 to briefly describe the topic of any question of the questionnaire. •    We improved the clinical relevance section of the Conclusions, reporting clinical implications of our study results and future perspective of treatment and protocols.